# Molecular Docking of Endolysins for Studying Peptidoglycan Binding Mechanism

**DOI:** 10.3390/molecules29225386

**Published:** 2024-11-15

**Authors:** Arina G. Arakelian, Gennady N. Chuev, Timur V. Mamedov

**Affiliations:** Institute of Theoretical and Experimental Biophysics, RAS, Institutskaya ul., 3, 142290 Pushchino, Moscow Oblast, Russia; genchuev@rambler.ru (G.N.C.);

**Keywords:** endolysin, peptidoglycan, cell wall, molecular docking, Autodock Vina, 3D-RISM

## Abstract

Endolysins of bacteriophages, which degrade the bacterial cell wall peptidoglycan, are applicable in many industries to deal with biofilms and bacterial infections. While multi-domain endolysins have both enzymatically active and cell wall-binding domains, single-domain endolysins consist only of an enzymatically active domain, and their mechanism of peptidoglycan binding remains unexplored, for this is a challenging task experimentally. This research aimed to explore the binding mechanism of endolysins using computational approaches, namely molecular docking and bioinformatical tools, and analyze the performance of these approaches. The docking engine Autodock Vina 1.1.2 and the 3D-RISM module of AmberTools 24 were studied in the current work and used for receptor–ligand affinity and binding energy calculations, respectively. Two possible mechanisms of single-domain endolysin–ligand binding were predicted by Autodock Vina and verified by the 3D-RISM. As a result, the previously obtained experimental results on peptidoglycan binding of the isolated gamma phage endolysin PlyG enzymatically active domain were supported by molecular docking. Both methods predicted that single-domain endolysins are able to bind peptidoglycan, with Autodock Vina being able to give accurate numerical estimates of protein–ligand affinities and 3D-RISM providing comparative values.

## 1. Introduction

Endolysins and drugs based on them [1,2,3] are extremely effective antimicrobial agents that have found application in many industries. First, endolysins have the ability to destroy biofilms, including antibiotic-resistant bacterial strains [4,5], and thus are applicable in biotechnology to control the growth of microorganisms as well as in the anti-microbial treatment of skin and work surfaces. In the food industry, endolysins can be used as safe and natural food preservatives [6,7,8,9,10,11].

Furthermore, when used in agriculture, endolysins can protect plants and animals from bacterial diseases. Transgenic tomato plants with the endolysin of the CMP1 phage, resistant to *Clavibacter michiganensis* [12], which causes anthracnose, and transgenic potatoes expressing T4 lysozyme [13,14] and resistant to *Pectobacterium carotovora* have been successfully created.

The effectiveness of endolysins against streptococci also makes it one of the means of treating mastitis in cows [15].

Naturally, endolysins are produced in the life cycle of bacteriophages (bacterial viruses)—obligate parasites that reproduce in host bacteria [16]. In order to digest the bacterial cell wall, bacteriophages have special proteins called lytic enzymes, or lysins [17]. The action of lysins results in the destruction of the bacterial cell wall and the release of phage progeny into the extracellular space [18]. One of these lytic enzymes, endolysin, is necessary at the stage of phage progeny release from the cell to destroy peptidoglycan by breaking peptide or glycosidic bonds [19].

The main component of the membranes of all bacterial cells is peptidoglycan, which is necessary for stabilizing cell membranes under conditions of high internal osmotic pressure [20]. Bacterial peptidoglycan is a complex polymeric structure. It has a carbohydrate backbone consisting of alternating N-acetylglucosamine and N-acetylmuramic acid residues linked by β(14)-glycosidic bonds, which is further “reinforced” by peptide subunits, often linked by interpeptide bridges. The peptide is linked to the N-acetylmuramic acid residue by an amide bond [21,22]. The composition of the peptide subunit, as well as the structure of the interpeptide bridges, varies among different bacterial strains, and these differences form the basis for the classification of peptidoglycans [23]. Types A and B differ in the position of the interpeptide bridge and the amino acid at the N-terminus of the peptide linked to the N-acetylmuramic acid residue. Peptidoglycans also differ in the structure of the interpeptide bridge and the type of amino acid at the position of the bond between the peptide subunits [24]. In addition to this, the main difference between the peptidoglycans of Gram-positive and Gram-negative bacteria is the thickness of the layers surrounding the plasma membrane. While the peptidoglycan of Gram-negative bacteria is only a few nanometers thick and contains one to several layers, Gram-positive peptidoglycan is 30–100 nm thick and contains multiple layers [25,26].

Differences in the structure of peptidoglycan in Gram-negative and Gram-positive bacteria are reflected in the pronounced functional and structural diversity of phage-encoded endolysins [27,28,29]. Endolysins specific to Gram-positive bacteria are modular proteins with two or more separate domains: EAD, enzymatically active domain (responsible for cleavage of peptide glycan), and one or more CBD(s)—cell wall binding domain(s) [30,31,32]. Endolysins of phages infecting Gram-negative bacteria are usually small globular proteins (15–20 kDa) and only have a catalytic domain [26,31,32,33,34]; however, there are also multidomain endolysins of Gram-negative bacteria [35,36,37,38].

Despite the existence of CBDs characteristic of the structures of various endolysins [39] and the establishment of common patterns responsible for the ability of EAD to bind peptidoglycan (namely, the presence of positively charged catalytic domain) [40], the mechanism of peptidoglycan binding by single-domain endolysins remains unexplored. Furthermore, experiments on the co-crystallization of endolysins with parts of bacterial cell walls do not always lead to interpretable results [41]. Therefore, alternative approaches to studying the mechanism of bacterial peptidoglycan binding by endolysins should be analyzed.

Molecular docking has emerged as a crucial tool in computational chemistry and structural biology, offering valuable insights into protein–ligand interactions [42,43,44,45]. There are two distinct approaches used in molecular docking simulations: rigid body docking and flexible docking. Rigid body docking assumes that both the receptor (protein) and ligand (small molecule or peptide) maintain fixed conformations throughout the docking process, simplifying the simulation by treating molecules as static entities [46,47]. Flexible docking acknowledges that both the receptor and ligand can change shape during the interaction, allowing for more realistic simulations of protein–ligand complexes [48,49]. Currently, there exists a number of academic (LeDock, rDock, AutoDock Vina, AutoDock (PSO), UCSF DOCK, and Au-toDock (LGA)) and commercial (GOLD, Glide (XP), Glide (SP), Surflex-Dock, LigandFit, and MOE-Dock) programs predicting receptor-ligand poses with an RMSD less than 2 Å between the top-scored pose and the native pose [50]. Among them, the academic program Autodock Vina, implementing a knowledge-based scoring function with a Monte Carlo sampling technique and the Broyden–Fletcher–Goldfarb–Shanno (BFGS) method for local optimization [51], is known for high prediction accuracy combined with a short simulation time.

Another possible approach to studying the thermodynamic properties of molecules in solution, including protein–ligand binding energies, relies on the three-dimensional reference interaction site model (3D-RISM)—an approximation to the molecular Ornstein-Zernike equation, utilizing a simulation force field and water model [52,53,54,55]. As with all theories based on the Ornstein–Zernike equation, solving the 3D-RISM equation requires a second equation, known as the closure relationship, which is necessarily approximate [56]. Among other existing closure relationships, the Kovalenko–Hirata (KH) closure approximation, a further extension to the KH closure approximation, and the partial series expansion of order n (PSE-n) have been shown to be successful and accurate for biomolecules in water [57,58,59,60,61,62,63,64,65]. Consequently, 3D-RISM-based calculations are known to produce values that are in disagreement with other empirical approaches and are recommended to be used in combination with other methods [66].

Current research relies on modeling methods that implement in silico molecular docking, exploring the Autodock Vina program together with the 3D-RISM approach to protein–ligand binding energy calculations. The structures of endolysins and bacterial peptidoglycan fragments are chosen to enable the comprehensive study of the peptidoglycan binding mechanism. This gives insights into the nature of amino acid residues involved in cell wall binding and suggests whether single-domain endolysins can be considered equivalent to the isolated EADs of multi-domain endolysins.

## 2. Results

### 2.1. Net Charge Calculation

Estimating the net charge of proteins at pH = 7.4 and isoelectric point values in Prot pi (Table 1), it appears that all four protein structures have a positive net charge, and therefore, it is theoretically possible for both EndoT5 and EAD of PlyG to bind bacterial peptidoglycan. The metal ions present in the structures are not included in the current calculation because the charge of metalloproteins is recommended to be measured directly due to differences between predicted and experimental values [67].

### 2.2. Autodock Vina Molecular Docking

Corresponding protein–ligand binding affinities, as calculated in Autodock Vina for each pair of protein and ligand, are shown in Table 2.

### 2.3. Binding Energies Calculations with 3D-RISM

To compare the results obtained with different bioinformatical methods, the AmberTools 3D-RISM package was used for binding energy calculations for four protein structures bound to NAM (Table 3).

As shown in Table 3, EndoT5-Zn^2+^ has the strongest affinity for NAM, with EndoT5-Zn^2+^-Ca^2+^ also being able to bind to this ligand, supporting the results obtained in Autodock Vina. The 3D-RISM approach predicted binding to NAM to be a highly energetically unfavorable process for both domains of PlyG.

Autodock Vina molecular docking offered two possible peptidoglycan-binding mechanisms for single-domain EndoT5. The first pattern involves only the N-terminus of the protein (Figure 1), and the second pattern, in addition to this, involves a part of the α-helix and a flexible linker formed by R26-Y33 residues (Figure 2), thus placing the ligand in a “cavity”. In cases with negative binding affinities (as shown in Table 2), when binding was observed, the second pattern of binding was characteristic of EndoT5-Zn^2+^, and the second pattern of binding was characteristic of EndoT5-Zn^2+^/Ca^2+^ in every case except for binding to CT1103206303 and CT1080279170 ligands; in this case, N-terminus binding of EndoT5-Zn^2+^ and “cavity” binding of EndoT5-Zn^2+^/Ca^2+^ were observed.

PlyG CBD was able to bind to different fragments of peptidoglycan, including peptidoglycan monomer (NAG-NAM-L-alanyl-D-isoglutaminyl-meso-diaminopimelic acid-D-alanyl-D-alanine), while PlyG EAD did not bind to any of the selected fragments except for muramyl pentapeptide (Figure 3).

## 3. Discussion

Autodock Vina molecular docking shows that for single-domain endolysins and separated EAD of multi-domain endolysin, ligand binding occurs on the side of the molecule opposite the protein’s active center (containing catalytic Zn^2+^ in both cases). However, the binding of separated EAD is much more specific: PlyG EAD was able to bind only to the muramyl pentapeptide (NAM-L-alanyl-D-isoglutaminyl-meso-diaminopimelic acid-D-alanyl-D-alanine). The ability of PlyG EAD to bind peptidoglycan has been previously supported by experimental data based on X-ray crystallography [40].

PlyG CBD was the only protein able to bind to peptidoglycan monomer, NAG-NAM-L-alanyl-D-isoglutaminyl-meso-diaminopimelic acid-D-alanyl-D-alanine, the major cell wall compound of *B. anthracis* [68], cell host of PlyG-producing *Bacillus phage gamma*. Meanwhile, the only fragment PlyG EAD was able to bind to was muramyl pentapeptide, NAG-NAM-L-alanyl-D-isoglutaminyl-meso-diaminopimelic acid-D-alanyl-D-alanine.

There are two predicted mechanisms of single-domain endolysin-peptidoglycan binding (Figure 4): N-terminal binding and “cavity” binding.

N-terminal binding mainly involves the exposed aromatic side chain of F3 and results in obstructing the bond cleaved by EndoT5; “cavity” binding involving R26-Y33 side chains results in exposing the bond cleaved by EndoT5.

The necessity of the involvement of single-domain Gram-negative EndoT5 N-terminus in binding peptidoglycan agrees with the existing results obtained during the analysis of structures available in the National Center for Biotechnology Information (NCBI) database [69]. According to this analysis, endolysins of Gram-negative bacteria predominantly have single-domain and higher hydrophobicity at the N-terminus, which makes this terminus a better candidate for substrate binding. This is not the case for PlyG EAD, which, being part of the Gram-positive bacteria endolysin, does not utilize the N-terminus for binding.

## 4. Materials and Methods

### 4.1. Endolysins and Fragments of Peptidoglycan In Silico Preparation

We selected four structures, all of which were obtained by solution NMR and deposited into the Protein Data Bank, as endolysin structures for this study: single-domain endolysins bacteriophage T5 L-alanoyl-D-glutamate peptidase Zn^2+^ form (Endo T5-Zn^2+^, PDB ID: 2MXZ) [70,71], bacteriophage T5 L-alanoyl-D-glutamate peptidase Zn^2+^/Ca^2+^ form (Endo T5-Zn^2+^/Ca^2+^, PDB ID: 8P3A) (Figure 5), and multi-domain hydrolase PlyG [72], represented by separate EAD (PDB ID:2L47) and CBD (PDB ID:2L48) structures (Figure 6). Prior to using these structures as receptors in Autodock Vina 1.1.2, we converted PDB files to PDBQT format using the Python-based software environment ADFRsuite 1.0. We estimated the net charge of proteins at pH = 7.4 in the bioinformatic toolbox Prot pi (https://www.protpi.ch/, accessed on 13 September 2024).

We obtained three-dimensional PDB structures of ligands, representing different parts of bacterial cell wall peptidoglycan, from the PubChem and Chemical Compounds Deep Profiling Services (CC-DPS) databases (https://pubchem.ncbi.nlm.nih.gov/, accessed on 8 August 2024, and https://www.cc-dps.com/, accessed on 8 August 2024, respectively). For the current study, we selected ligands that included peptidoglycan fragments typical of various bacterial species. Our goal was to represent comprehensively different types of peptidoglycan structures.

The list of ligand includes N-acetylglucosamine (NAG, PubChem CID 24139) (Figure 7A), N-acetylmuramic acid (NAM, PubChem CID 5462244) (Figure 7B), NAG-NAM dimer (PubChem CID 72210857) (Figure 7C), NAM-L-alanine (PubChem CID 10970945) (Figure 7D), NAM-L-alanyl-D-isoglutamine (PubChem CID 451714) (Figure 7E), NAM-L-Ala-ϒ-D-Glu-L-Lys-D-Ala-D-Ala (CC-DPS CT1103206303) (Figure 7F), muramyl pentapeptide (CC-DPS CT1080279170) (Figure 7G), peptidoglycan monomer (CC-DPS CT1079218991) (Figure 7H), and pentaglycine (PubChem CID 81537) (Figure 7I). Finally, we converted PDB files to PDBQT format using the Python package Meeko v0.4.0 in order to use the aforementioned structures as ligand structures in Autodock Vina.

### 4.2. Molecular Docking and Binding Energy Calculations

We performed blind (global) molecular docking in Autodock Vina (https://vina.scripps.edu/). To prepare the GPF file for AutoGrid4, we used the prepare_gpf.py command line tool with the grid automatically centered around the ligand. Autodock Vina was run using the AutoDock4 forcefield. Running AutoDock Vina resulted in obtaining PDBQT files containing mutual arrangements for each pair of protein and ligand found during molecular docking and the affinity of the aforementioned protein to said ligand.

Using the molecular visualization system PyMOL (https://pymol.org/), we converted PDBQT files containing the structures of proteins with docked ligands into PDB format in order to obtain three structures: protein in ligand-binding conformation, docked ligand, and the protein with docked ligand. A further run of the 3D-RISM AmberTools package (https://ambermd.org/AmberTools.php) allowed us to calculate the solvation free energy of each structure [73,74,75]. We executed separate calculations for KH and PSE2 closures, with the minimum distance between the solute and the edge of the solvation box equal to 15 Å and the spacing between grid points equal to 0.4 Å. We computed binding energies as follows:(1)$$E_{bind}=E_{complex}-E_{protein}-E_{ligand},$$
where $E_{complex}$ is the solvation free energy of protein with a docked ligand, $E_{protein}$ is the solvation free energy of the protein in ligand-binding conformation, and $E_{ligand}$ is the solvation free energy of the docked ligand [76].

We obtained images showing proteins with docked ligands and the distances in intermolecular interactions using the molecular visualization system PyMOL.

## 5. Conclusions

Modeling methods implementing in silico molecular docking, namely Autodock Vina and 3D-RISM, proved to be useful bioinformatics tools, allowing us to study the binding of single-domain endolysins to different ligands representing part of bacterial wall peptidoglycan. Single-domain peptidoglycans were predicted to bind peptidoglycan with specific parts of their structure located on the side opposite to the protein’s active center, including the exposed aromatic side chain on the N-terminus and, occasionally, cavities formed by α-helices and flexible linkers, thus resulting in two theoretical mechanisms of binding: N-terminal binding and “cavity” binding. The pattern of “cavity” binding of peptidoglycan was common for both single-domain endolysins and separate EADs of multi-domain endolysins, with the latter characterized by more ligand-specific binding.

It can be concluded that Autodock Vina gives accurate numerical estimates of protein–ligand affinities, as it is in agreement with experimental data, while 3D-RISM remains an approach suitable for obtaining comparative values and analyzing common trends of thermodynamic properties differing among protein–ligand pairs. Therefore, both methods are recommended to be used in combination.

## Figures and Tables

**Figure 1 molecules-29-05386-f001:**
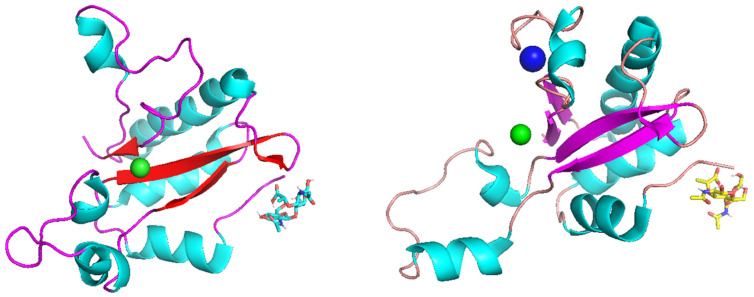
N-terminus binding of endolysins Endo T5-Zn^2+^ (**left**) and Endo T5-Zn^2+^/Ca^2+^ (**right**) to the NAG-NAM ligand, with Zn^2+^ ion represented by green spheres and Ca^2+^ represented by a blue sphere.

**Figure 2 molecules-29-05386-f002:**
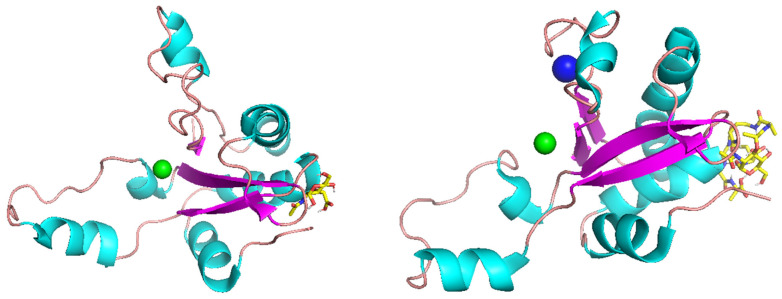
“Cavity” binding of endolysins Endo T5-Zn^2+^ (**left**) and Endo T5-Zn^2+^/Ca^2+^ (**right**) to NAM and CC-DPS CT1103206303 ligands, respectively, with Zn^2+^ ion represented by green spheres and Ca^2+^ represented by a blue sphere.

**Figure 3 molecules-29-05386-f003:**
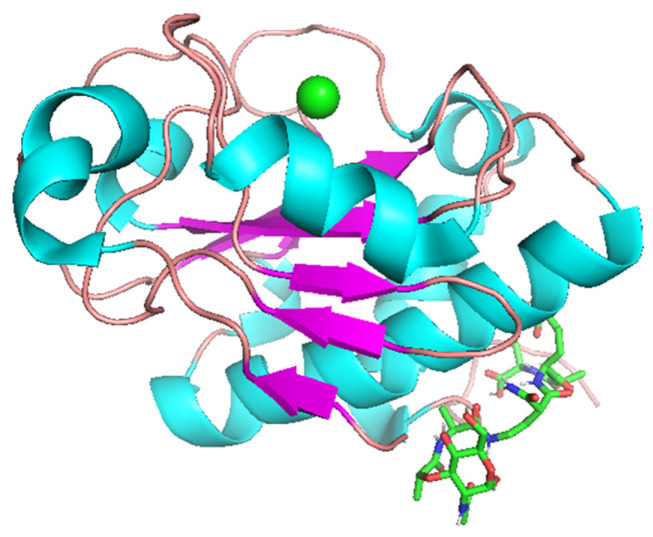
PlyG EAD bound to muramyl pentapeptide, with Zn^2+^ ion represented by a green sphere.

**Figure 4 molecules-29-05386-f004:**
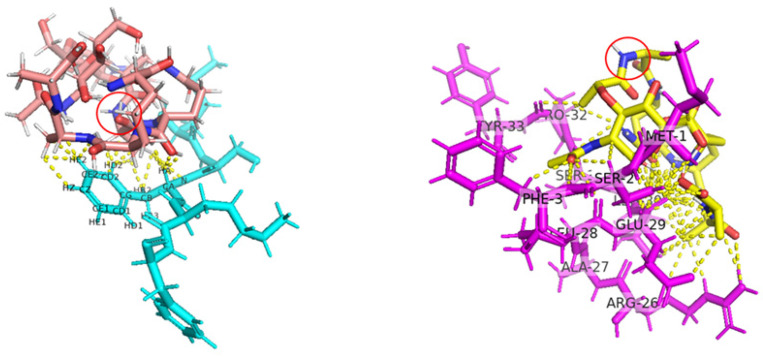
EndoT5 Zn^2+^-Ca^2+^ binding peptidoglycan monomer (**left**) and CC-DPS CT1103206303 (**right**) as examples of N-terminal and “cavity” binding processes, respectively. The yellow dotted line represents distances between atoms that are less or equal to 2.5 Å, corresponding to hydrogen bonds; atoms of F3 (**left**) and amino acid residues participating in ligand binding (**right**) are signed. Red circles denote the bond of the peptidoglycan that is cleaved by EndoT5.

**Figure 5 molecules-29-05386-f005:**
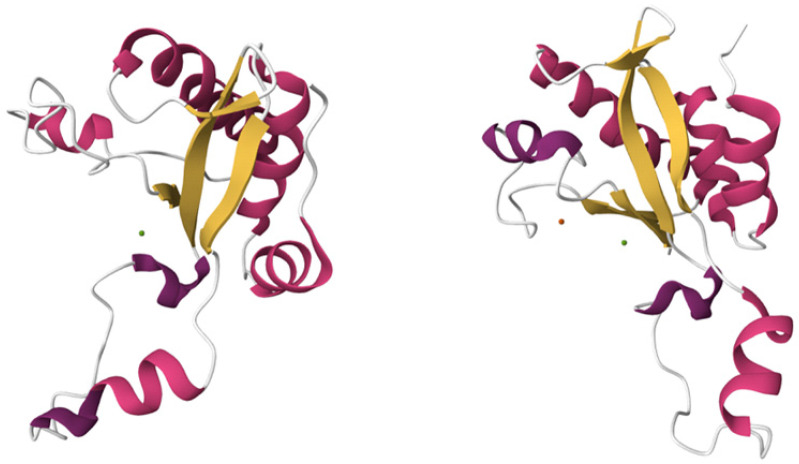
Single-domain endolysins Endo T5-Zn^2+^ (**left**) and Endo T5-Zn^2+^/Ca^2+^ (**right**) with Zn^2+^ ion represented by green spheres and Ca^2+^ represented by an orange sphere.

**Figure 6 molecules-29-05386-f006:**
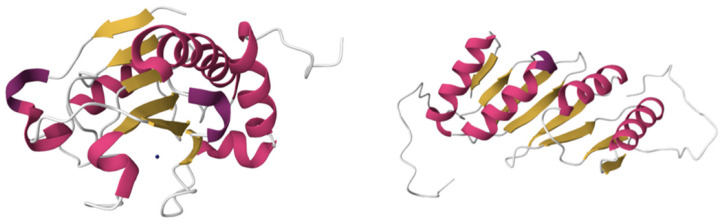
Multi-domain endolysin PlyG divided into EAD (**left**) and CBD (**right**) with the Zn^2+^ ion represented by a blue sphere.

**Figure 7 molecules-29-05386-f007:**
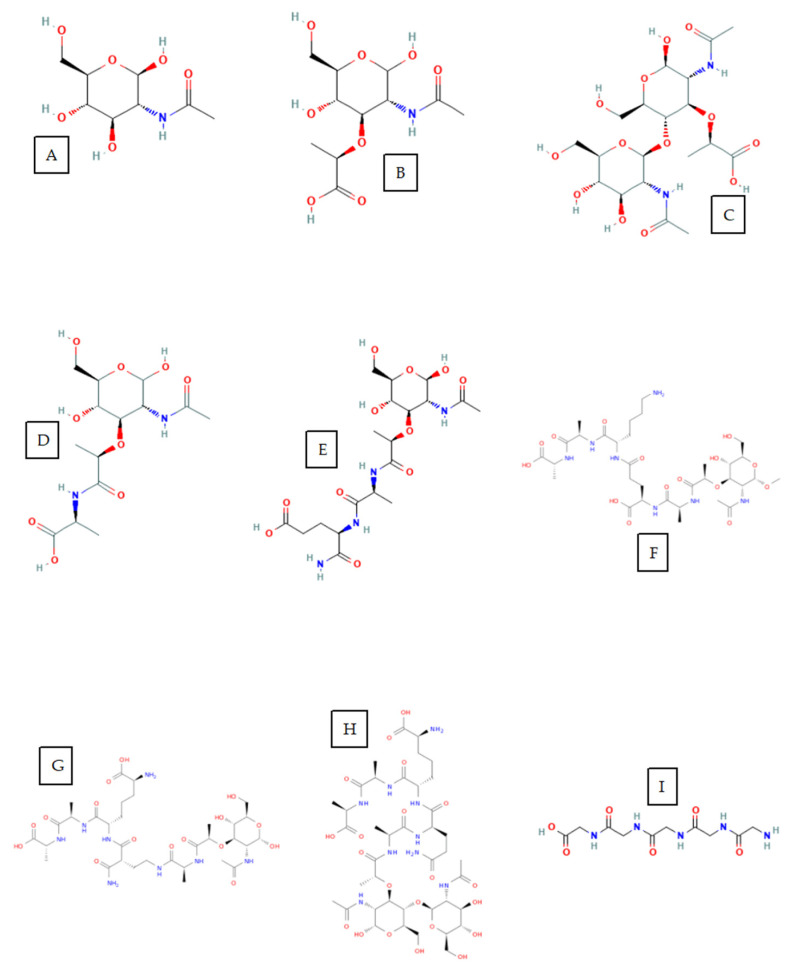
Ligand structures: NAG (**A**), NAM (**B**), NAG-NAM (**C**), NAM-L-Ala (**D**), NAM-L-Ala-ϒ-D-Glu (**E**), CC-DPS CT1103206303 (**F**), muramyl pentapeptide (**G**), peptidoglycan monomer (**H**), and pentaglycine (**I**).

**Table 1 molecules-29-05386-t001:** Physicochemical parameters of proteins.

Parameter	EndoT5	PlyG EAD	PlyG CBD
Isoelectric point	7.823	8.125	8.781
Net charge at pH = 7.4	+0.356	+1.918	+1.516

**Table 2 molecules-29-05386-t002:** Protein–ligand binding affinities calculated using Autodock Vina, kcal/mol.

↓Protein →Ligand	NAG	NAM	NAG-NAM	NAM-L-Ala	NAM-L-Ala-ϒ-D-Glu	CC-DPS CT1103206303	CC-DPS CT1080279170	CC-DPS CT1079218991	Penta-Glycine
EndoT5-Zn^2+^	1.786	−5.403	−3.737	−4.276	−3.022	−0.2375	1.002	1.937	−3.798
EndoT5-Zn^2+^-Ca^2+^	1.788	−2.773	−1.674	−2.685	−0.9778	−1.075	−1.897	0.551	−2.112
PlyG CBD	1.788	−2.588	−2.701	−3.229	−1.730	0.586	1.191	−2.640	−2.685
PlyG EAD	1.788	12.52	416	111.6	277.9	2709	−3.000	4105	31.95

**Table 3 molecules-29-05386-t003:** Protein–ligand binding energies calculated using 3D-RISM, kcal/mol.

Protein	KH Closure	PSE2 Closure
EndoT5-Zn^2+^	−5.743	−12.20
EndoT5-Zn^2+^-Ca^2+^	−0.107	−4.312
PlyG CBD	26.20	27.76
PlyG EAD	73.54	74.23

## Data Availability

The original contributions presented in this study are included in the article. Further inquiries can be directed to the corresponding author.

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
