# Peer review of "Molecular Docking of Endolysins for Studying Peptidoglycan Binding Mechanism"

_molecules, 2024, doi:10.3390/molecules29225386_

Round 1

Reviewer 1 Report

Comments and Suggestions for Authors

The manuscript presents interesting results and it is recommended for publication pending revisions.

-Regarding the introduction, the authors included literature from various years; however, the most recent cited article is from 2023. The authors should include more recent findings (2024 publications) in the introduction to provide readers with an up-to-date context in the research area.

-In the materials and methods section on molecular docking (4.2), the authors provided limited details about the docking procedures. They should clarify in the manuscript whether blind docking was performed across the entire macromolecule.

-Concerning molecular docking, the authors should specify the criteria for selecting conformations. Beyond selecting minimum-energy conformations, clustering statistics should be considered. The authors need to clarify these aspects in the manuscript.

-Interaction energies were calculated using the 3D-RISM AmberTools package. Based on the manuscript, partial charges were seemingly obtained directly from the docking software. For more reliable energy values, charge calculations should ideally be performed using ab initio methods. The authors should address these points.

-In the discussion, the authors provided a very brief analysis of molecular interactions, which is a central topic of the manuscript. They should elaborate on and discuss these interactions in more detail, particularly specific interactions such as pi-pi, cation-pi, and hydrogen bonds (e.g., 10.3390/molecules25122841 and 10.3390/molecules28196891).

Author Response

Thank you very much for reviewing our manuscript. Below, you will find detailed responses addressing each point raised, along with the corresponding corrections (shown in red) in the resubmitted files.

Comment 1: Regarding the introduction, the authors included literature from various years; however, the most recent cited article is from 2023. The authors should include more recent findings (2024 publications) in the introduction to provide readers with an up-to-date context in the research area.
Response 1: Thank you for this comment. In light of your suggestion, we have updated our references to incorporate publications that are more recent. Specifically, we have changed references №4 and №5, which support the assertion that endolysins can effectively disrupt biofilms, including those formed by antibiotic-resistant bacterial strains.
These newly introduced references can be found on page 9, lines 298-303 of the current document.

Comment 2: In the materials and methods section on molecular docking (4.2), the authors provided limited details about the docking procedures. They should clarify in the manuscript whether blind docking was performed across the entire macromolecule.
Response 2: We acknowledge the significance of this matter and appreciate your attentiveness. We have made modifications to line 232 on page 7: “We performed blind (global) molecular docking in Autodock Vina (https://vina.scripps.edu/)”

Comment 3: Concerning molecular docking, the authors should specify the criteria for selecting conformations. Beyond selecting minimum-energy conformations, clustering statistics should be considered. The authors need to clarify these aspects in the manuscript.
Response 3: In our assessment, we believe that all necessary criteria for selecting conformations have been fulfilled.
There exist several sets of recommendations regarding the selection of protein conformations, including those found in the following publications:

https://doi.org/10.1021/ci9003943

https://doi.org/10.1038/s41596-021-00597-z

These guidelines primarily focus on crystal structures and emphasize the importance of choosing a protein conformation that is bound to a substrate or demonstrates structural similarity to a bound conformation. In contrast, our research is based on structures of catalytically active proteins obtained by NMR spectroscopy in solution. A set of these structures, representing the lowest energy conformations, was deposited into the Protein Data Bank.
AutoDock was utilized to select the optimal protein-ligand pair from this dataset. Consistent with the recommendations outlined in https://doi.org/10.1038/s41596-021-00597-z, protonation was incorporated into the AutoDock protocol, resulting in the automatic consideration of protonated states during the docking process.

Comment 4: Interaction energies were calculated using the 3D-RISM AmberTools package. Based on the manuscript, partial charges were seemingly obtained directly from the docking software. For more reliable energy values, charge calculations should ideally be performed using ab initio methods. The authors should address these points.
Response 4: We have conducted previous 3D RISM calculations using both classical force field charges and those derived from quantum mechanical (QM) calculations, https://doi.org/10.1021/ct2009297.
Our findings suggest that the incorporation of QM partial charges does not enhance the accuracy of the calculations; instead, it appears to increase the discrepancy between experimental and computed data.
This discrepancy may be attributed to the fact that 3D RISM calculations are based on a nonzero repulsion radius for hydroxyl hydrogen atoms. In contrast, this radius is optimized specifically for classical force field charges.

Comment 5: In the discussion, the authors provided a very brief analysis of molecular interactions, which is a central topic of the manuscript. They should elaborate on and discuss these interactions in more detail, particularly specific interactions such as pi-pi, cation-pi, and hydrogen bonds (e.g., 10.3390/molecules25122841 and 10.3390/molecules28196891).
Response 5: Thank you for bringing it to our notice. We have added text displayed in red to the description of Figure 4 (located on page 5, lines 173-174). Consequently, the description now states the following: Yellow dotted line represents distances between atoms that are less or equal to 2.5 Å, corresponding to hydrogen bonds; atoms of F3 (left) and amino acid residues participating in ligand binding (right) are signed.
Hydrogen bonds typically exhibit lengths ranging from approximately 1.5 to 2.5 Å. Given that docking simulation in question incorporates hydrogen bonding interactions, we initially assumed that specifying the distance would suffice for understanding these interactions. However, we now recognize the need for explicit clarification.
With current work being devoted to identifying ligands that bind with proteins and the corresponding amino acid residues involved in binding, we believe that in-depth analysis of molecular interactions presents an intriguing area for further research.

Respectfully, 
Arina Arakelian

Reviewer 2 Report

Comments and Suggestions for Authors

See the file attached

Author Response

Thank you very much for reviewing our manuscript. Below, you will find detailed responses addressing each point raised, along with the corresponding corrections (shown in red) in the resubmitted files.

Comment 1: In the abstract (line 18) the acronym PlyC is used without introducing (at least for the first time the full name)
Response 1: We appreciate your attention to this detail. Although PlyG is the name of endolysin (as referenced in doi:10.1016/j.virol.2014.11.003), we have decided that incorporating the full name, gamma phage endolysin PlyG, as written in red on page 1, line 18, will enhance the clarity of our current work.

Comment 2: At page 4 the author reports: “ PlyG CBD was able to bind to different fragments of peptidoglycan, including peptidoglycan monomer (NAG-NAM-L-alanyl-D-isoglutaminyl-mesodiaminopimelic acid-D-alanyl-D-alanine) while PlyG EAD did not bind to any of selected fragments except for muramyl pentapeptide (Figure 3).” An explanation of the reason of the process can help the reader to better understand the behavior.
Response 2: Unfortunately, it is not currently feasible to draw any definitive conclusions regarding the reasons for the said behavior primarily due to the lack of data on the three-dimensional structure of peptidoglycan. Our analysis is limited to comparing the behaviors of two distinct domains, which exhibit differences that may be attributed to variations in their ability to penetrate pores of different sizes or bind at different angles.
Previously we have stated in “Discussion” section the following: “PlyG-CBD was the only protein able to bind to peptidoglycan monomer, NAG-NAM-L-alanyl-D-isoglutaminyl-meso-diaminopimelic acid-D-alanyl-D-alanine, the major cell wall compound of B. anthracis [68], cell host of PlyG-producing Bacillus phage gamma”.
Adding to this  “Meanwhile, the only fragment PlyG-EAD was able to bind to was muramyl pentapeptide, NAG-NAM-L-alanyl-D-isoglutaminyl-meso-diaminopimelic acid-D-alanyl-D-alanine.”  (page 6, lines 179 – 180) allows us to compare behavior of two PlyG domains.

Comment 3: Is it possible to have an estimation of the uncertainties of the values of energy reported in table 3?
Response 3: 
It is not possible because we have solved the 3D-RISM equation using two specific closure relationships. In order to make conclusions on the usability of these closure relationships for certain solvent-solute systems, it is possible to compare our results with experimental data; however, working on 3D-RISM theory lies beyond the scope of this work.

Comment 4: Maybe the author can add a few sentences to better explain what are the two values reported in table 3 (and why are they different from each other)
Response 4: 3D-RISM generates correlation functions, showing how solvent molecules interact with solutes or larger molecular structures. A crucial component of 3D-RISM is a mathematical relationship that completes the equations describing these interactions – closure relationship. This relationship must accurately capture both the long-term behavior and close-range characteristics of these interactions, and closure relationship is always approximate.
According to lines 100 – 103 on page 3, “Among other existing closure relationships, Kovalenko-Hirata (KH) closure approximation, and a further extension to the KH closure approximation, the partial series expansion of order n (PSE-n), have been shown to be successful and accurate for biomolecules in water [57 – 65].”, and our goal was to show that “3D-RISM remains an approach suitable for obtaining comparative values and analyzing common trends of thermodynamic properties differing among protein-ligand pairs”, as stated in lines 269 – 271 on page 9. This conclusion is common for both sets of values, i.e., for both closure relationships, and characterizes the method itself.

Comment 5: The discussion in section 3 can be increased to have a better picture of the importance of the obtained results
Response 5: 
We agree with this comment. Please find this section extended with the following information (page 6, lines 186 – 192):
The necessity of involvement of single-domain Gram-negative EndoT5 N-terminus in binding peptidoglycan agrees with existing results obtained during the analysis of structures available in the National Center for Biotechnology Information (NCBI) database [69]. According to this analysis, endolysins of Gram-negative bacteria predominantly have single domain and higher hydrophobicity at the N-terminus, which make this terminus a better candidate for substrate binding. This is not the case for PlyG-EAD, which, being part of Gram-positive bacteria endolysin, does not utilize the N-terminus for binding.

  1. Vazquez, R., Garcia, E., Garcia, P. Sequence-Function Relationships in Phage-Encoded Bacterial Cell Wall Lytic Enzymes and Their Implications for Phage-Derived Product Design. J. Virol.. 2021, 95(14), e0032121.

Comment 6: In section 4.2 (line 240) is reported “with minimum distance between the solute and the edge of the solvation box equal to 15 Å and spacing between grid points equal to 0.4 Å.” Is there a particular reason for the choice of the parameters?
Response 6:
According to AmberTools Manual, section 7.2.3., page 119, these parameters are considered to be adequate for modelling solvation of the whole protein.

Respectfully, 
Arina Arakelian